# Association of Human Cytomegalovirus exposure with tuberculosis disease in South African adults with presumptive tuberculosis

Derrick Semugenze[1,2], Arthur Chiwaya[3], George William Kasule[4], James Sserubiri[4], Rose Nabatanzi[1], Byron W. P. Reeve[3†], Zaida Palmer[3], Hridesh Mishra[3,5], Achilles Katamba[6], Alberto García-Basteiro[7], Moses L. Joloba[1], Grant Theron[3], Frank Cobelens[2], Willy Ssengooba[4*]

1 Department of Immunology and Molecular Biology, Makerere University College of Health Sciences Kampala, Kampala, Uganda, 2 Department of Global Health and Amsterdam Institute for Global Health and Development, Amsterdam University Medical Centers Location University of Amsterdam, Amsterdam, The Netherlands, 3 Division of Molecular Biology and Human Genetics, Faculty of Medicine and Health Sciences, DSI-NRF Centre of Excellence for Biomedical Tuberculosis Research, South African Medical Research Council Centre for Tuberculosis Research, Stellenbosch University, Cape Town, South Africa, 4 Department of Medical Microbiology, Makerere University College of Health Sciences Kampala, Kampala, Uganda, 5 Sandra A. Rotman (SAR) Laboratories, Sandra Rotman Centre for Global Health, University Health Network-Toronto General Hospital, Toronto, Ontario, Canada, 6 Clinical Epidemiology & Biostatistics Unit, Department of Medicine, Makerere University College of Health Sciences, Kampala, Uganda, 7 Centro de Investigação em Saúde de Manhiça (CISM), Maputo, Mozambique, ISGlobal, Barcelona Centre for International Health Research, Hospital Clínic - Universitat de Barcelona, Barcelona, Spain

† Deceased.
* willyssengooba@gmail.com

## Abstract

Recent studies suggested that human cytomegalovirus (HCMV) exposure may increase tuberculosis (TB) disease risk. We assessed the association between active HCMV infection and recent HCMV exposure with tuberculosis (TB) disease among TB-presumptive South African adults. This was a cross-sectional case-control study that utilized stored plasma and serum samples collected from adults (≥18 years) with presumptive TB self-presenting to primary care clinics in in the Kraaifontein District in Cape Town, South Africa. This study analyzed TB Cases (n = 98) who were mycobacterial culture and or GeneXpert Ultra positive and controls (n = 199) who were frequency matched by HIV status. Current HCMV infection (including reactivation or reinfection) and recent infection were defined using qPCR and serology (IgM and IgG avidity ELISA), while HCMV DNAemia was defined by a positive qPCR result alone. In a logistic regression model adjusting for age, gender, HIV status and BMI, TB disease was associated with HCMV DNAemia [adjusted odds ratio (aOR) 4.99, 95%CI 1.63–16.99, p = 0.007]. A similar association was observed for current HCMV infection, whereas no association was found for recent HCMV infection. These results indicate that active HCMV replication although not frequent may impair immune

**Data availability statement:** All data generated or analyzed during this study are included in this published article and its Supporting information files.

**Funding:** This work was supported by the PreFIT project (EDCTP2 programme, grant agreement number RIA2018D-2509 to FC), Mr. Willem Bakhuys Roozeboomstichting (MrWBR-CMV&TB Substudy-FC, grant number 0229 to FC) and Saharan African Network for TB/HIV Research Excellence (SANTHE; https://www.santheafrica.org/) (Grant number INV-033558 to DS). The funders had no role in study design, data collection and analysis, decision to publish, or preparation of the manuscript.

**Competing interests:** The authors have declared that no competing interests exist.

response to TB disease, TB disease could be leading to HCMV replication or an underlying common factor reactivates HCMV and Mtb replication in this population.

## Introduction

About 10.7 million people developed tuberculosis (TB) disease in 2024 and the incidence rate is back to the pre-COVID-19 pandemic levels [1]. About 5–15% of the *Mycobacterium tuberculosis* (Mtb) latently infected individuals progress to active disease within their lifetime [2], but knowledge on the factors that lead to disease progression is incomplete. Exposure to human cytomegalovirus (HCMV), a herpesvirus that remains present lifelong after (re-)infection and can reactivate, has been suggested as one of such factors [3]. In a recent systematic review, all studies that investigated HCMV associated it with TB disease [4]. HCMV infection was unrelated to tuberculin skin test conversion [5], suggesting this association may reflect enhanced disease progression due to active HCMV infection. Between 60% and 90% of adults are estimated to be HCMV seropositive worldwide with highest prevalence in non-Caucasian populations and low socioeconomic settings [6]. HCMV modulates the host immunity, amongst others by causing change overtime in the T-cell repertoire through significant depletion of naïve T-cells and increased numbers of memory T cells, a phenomenon referred to as immunosenescence [7]. This might possibly reduce the ability of the host to mount an effective adaptive immune response against pathogens including Mtb; since T-cells are crucial for controlling Mtb infection this could lead to unstable control of Mtb and disease progression [8]. While other studies have looked at circulating anti-HCMV IgG levels, two studies in South Africa showed increased risk of TB disease among infants and children with HCMV shedding and HCMV-specific interferon-gamma responses, respectively [5,9]. These studies followed up infants from birth, and hence associated primary HCMV infection with primary TB disease [5]. Whether these findings can be extrapolated to adults is unknown. Over 90% of adults in developing countries are seropositive for HCMV [10] and this are likely to be reinfections or reactivations since the virus establishes latency after primary infection [11,12]. In TB endemic countries, adults are likely to have HCMV reactivation or even reinfection rather than primary disease [13]. We therefore set out to assess whether HCMV viremia and reactivation or reinfection is associated with TB disease in South African TB presumptive adults.

## Materials and methods

### Ethics statement

The mother study (BAR-TB Dx) had obtained Ethics approval (Ethics Reference No: N14/10/136) and all participants had provided written informed consent for their stored samples to be used for future analyses. We obtained ethical approval from the Makerere University College of Health Sciences School of Biomedical Sciences Research and Ethics Committee (Approval number: SBS-2023–402) and Uganda National Council for Science and Technology (Approval number: HS3255ES).

## Study design and participant flow

Adults (≥ 18 years) with presumptive pulmonary TB disease [14], not currently on treatment or within the previous two months were consecutively recruited between 06th February 2016 and 22nd February 2023 at Primary health care clinics in the Kraaifontein District in Cape Town, South Africa (Fig 1). De-identified demographic, clinical and laboratory data relevant to this study that was captured in REDCap [15] was downloaded on 12th June 2024. Participants without TB diagnosis and those with contaminated cultures without GeneXpert MTB/RIF Ultra results were excluded from subsequent analysis. Pulmonary tuberculosis (TB) cases were defined as individuals with a positive mycobacterial culture and/or GeneXpert MTB/RIF Ultra (Cepheid Sunnyvale, CA, United States) result. The sample size was based on the earlier study that was conducted in this population [16] from which the sampling frame of TB presumptive patients confirmed to have pulmonary TB, 34 individuals living with HIV and all the 64 TB cases without HIV were selected. We aimed to include twice as many TB cases without HIV infection; however, we included all available numbers in the sampling frame. Controls were randomly selected from presumptive pulmonary TB patients who tested negative for both mycobacterial culture and GeneXpert MTB/RIF Ultra. Controls were selected to achieve approximately a 2:1 control-to-case ratio and were frequency matched to TB cases by HIV status. This resulted in 71 controls living with HIV and 128 controls without HIV infection. We expected the sample size to allow us to find a 25% or larger prevalence of HCMV infection among patients diagnosed with TB as significantly (level 0.05) different from an HCMV prevalence among the control group of 5% (power 0.8), allowing for controlling for confounding variables. Both cases and controls were recruited from the Primary health care clinics.

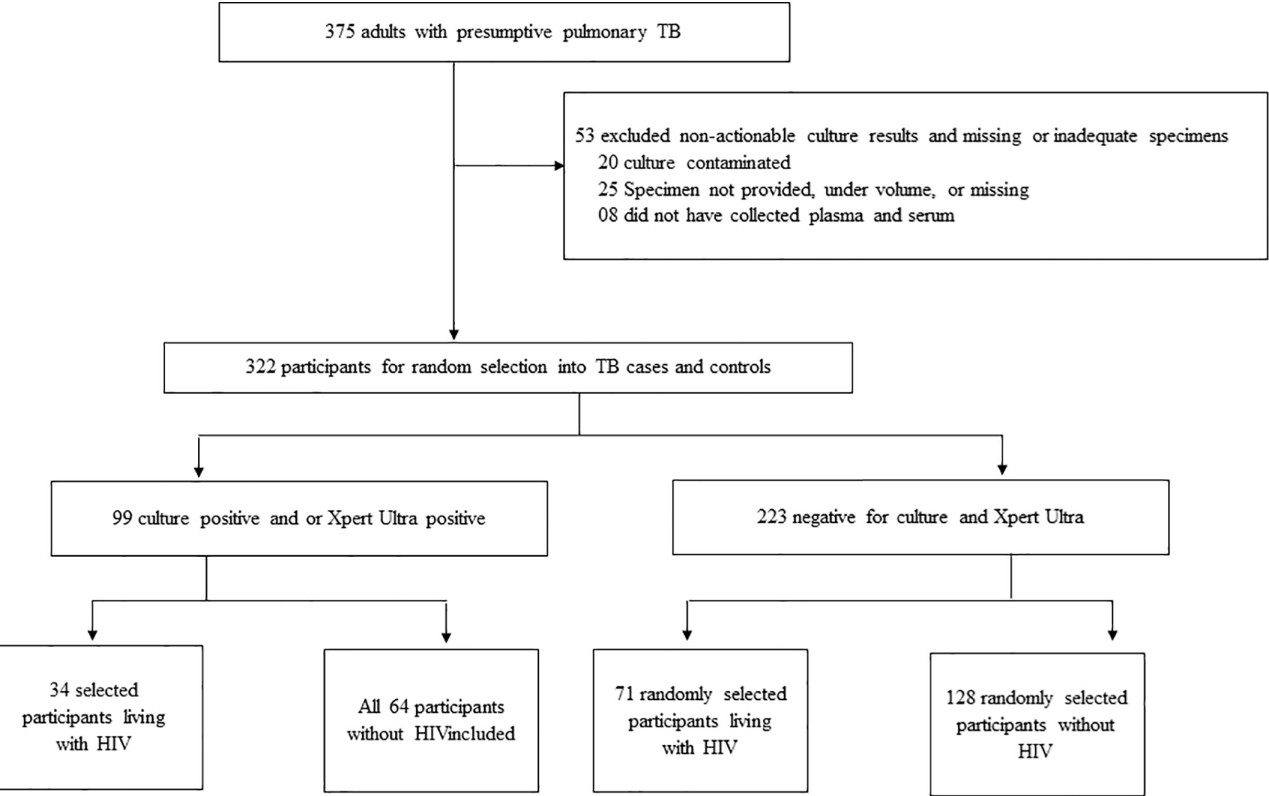

**Fig 1. Flowchart illustrating the selection and categorization process of study participants. TB; tuberculosis.**

## Laboratory procedures

Plasma and serum samples were initially stored at -80°C and shipped on dry ice to the Integrated Biorepository of H3Africa Uganda, Makerere University, where they were immediately stored at -80°C before the analysis.

**Real-time qPCR.** HCMV DNAemia was determined by extracting viral DNA from plasma using QIAamp DSP Virus Spin kit (Qiagen, Hilden, Germany) following the manufacturer's instructions [17]. Real-time qPCR was done using artus CMV RG PCR kit (Qiagen, Hilden, Germany) on the QuantStudio5 platform using the following parameters as stated by the manufacturer [18]: hot start enzyme activation at 95°C for 10 minutes, denaturation at 95°C for 15 seconds, annealing at 65°C for 30 seconds, and extension at 72°C for 20 seconds. Forty-five cycles were done during denaturation and annealing. This assay had a HCMV DNA limit of Detection (LoD) of 11.42 copies/ml in plasma. The HCMV DNAemia results were qualitatively analyzed (positive/negative).

**Detection of HMCV specific IgM antibodies.** Euroimmun HCMV ELISA Kits (Lübeck, Germany) with coated recombinant EI-p52 antigen were used according to the manufacturer`s instructions. The participant sera were diluted 1:101 before pipetting 100μl into the microtiter wells and incubated for 30 minutes. The wells were then washed. After that, 100μl of horseradish peroxidase-conjugated anti-human IgM antibodies were transferred into microtiter wells and still incubated for 30 minutes. The wells were then washed to remove the unbounded conjugate from immune complexes. The wells were incubated with TMB substrate for 15 minutes. The positive control was monitored for development of a blue color. 100μl of the stopping solution (sulfuric acid) was then added to stop the enzymatic reaction (the blue color turned to yellow). For measuring the optical density (OD) of the samples, the Biotek Synergy H1 Multi-Mode Microplate reader (Winooski, Vermont, USA) was used at 450 nm using 620nm as reference. Participants were considered seropositive when the ratio of extinction of the patient sample to that of the calibrator was ≥ 1.1.

**Determination of HMCV specific IgG avidity antibodies.** This was done to confirm diagnosis of primary HCMV infection [19,20]. High IgG avidity occurs 2–4 months after primary infection. The manufacturer`s procedure was followed, similar to the one described above for IgM antibody determination, but with running each sample in duplicate and an additional step of adding 100μl of Urea and Phosphate Buffered Saline (PBS) in each duplicate followed by incubation at room temperature for 10 minutes after the step of washing out the diluted sera from the microtiter wells. Participants were considered to have low and high avidity antibodies when the relative avidity index (RAI) was < 40% and >60% respectively.

## Statistical analysis

The raw data file, which included the supplementary dataset (S1 Data), was imported into R studio software (Version R.4.5.1) for data analysis. Sociodemographic and clinical characteristics of the participants were summarized using median for age and frequency tables for other variables. We determined crude odds ratios and then performed univariate logistic regression to assess the association between each independent variable and TB disease. A Directed Acyclic Graph for theoretical causal reasoning was drawn (Fig A in S1 Text) to categorize different variables in the dataset into potential confounders, mediators and effect modifiers for the association between HCMV DNAemia, primary infection, current or recent HCMV infection and TB disease based on the scientific knowledge and published literature, as guided by [21].

Our primary analysis investigated the association of TB disease with HCMV DNAemia. Potential confounders were included in multivariable logistic regression models for empirical data-driven validation along with age, gender and HIV status if they showed a difference of ≥10% between the crude odds ratio and adjusted odds ratio, a restriction added to prevent overfitting. Body Mass Index (BMI) was also examined for effect modification of the association between HCMV DNAemia and TB disease. All analyses used a p = 0.05 significance level. Hemoglobin attenuated the association after including it in the multivariable logistic regression model which prompted its examination in a multivariable mediation analysis estimating direct, indirect and total effect with nonparametric bootstrap confidence intervals (1000 simulations) using the mediation R package.

Our secondary analysis investigated the association of TB disease with past HCMV infection, primary HCMV infection, current HCMV infection (this may include new infection, reactivation or reinfection) and recent HCMV infection (which may include recent infection, reactivation or reinfection), for which four categories were created (Box).

> **Box: Categories of HCMV exposure applied in secondary analysis**
>
> **Primary HCMV infection**: *Participants with low anti-HCMV IgG avidity irrespective of HCMV DNAemia and anti-HCMV IgM results [19,20]*
>
> **Current HCMV infection**: *Participants with HCMV DNAemia and high anti-HCMV IgG avidity irrespective of anti-HCMV IgM results*
>
> **Recent HCMV infection**: *Participants without HCMV DNAemia, and positive for anti-HCMV IgM with high anti-HCMV IgG avidity*
>
> **Past HCMV infection**: *Participants without HCMV DNAemia who tested negative for anti-HCMV IgM and had high anti-HCMV IgG avidity*

The same procedure described above for HCMV DNAemia was used for identification of potential confounders and inclusion in the multivariable model.

Descriptive exploratory analyses were also conducted to assess the effect of HCMV viral load on several TB-related outcomes, including; Time to TB Positivity (TTP) in mycobacterial culture, Mtb culture results, and previous TB disease history. Finally, TB disease classification was examined by categorizing individuals into those with positive TB cultures without prior TB and those with positive cultures reporting previous TB.

## Results

### Description of study participants and characteristics of TB cases and controls

A total of 375 participants were screened to participate in this study. After the exclusion of 53 participants due to non-actionable culture results, missing samples, and collected samples with insufficient volumes, 322 were eligible to participate in the study. A total of 297 participants including 98 TB cases (34 PLWHIV and 64 non-HIV infected) and 199 controls (71 PLWHIV and 128 non-HIV infected) were selected for this study (Fig 1).

The median age of all the participants was 37 years (interquartile range, IQR 29–47). Detection of HCMV DNAemia was observed in 12 (12.2%) of TB cases and 9 (4.5%) controls (Table 1). The age distribution differed significantly across groups (p=0.005). Among TB cases, HCMV positivity was more frequent in individuals aged 36–50 years (58.3%), whereas HCMV-negative TB cases were predominantly aged 18–35 years (55.8%). A similar pattern was observed among controls, where 77.8% of HCMV-positive individuals were aged 36–50 years, compared with 37.9% of HCMV-negative controls aged 18–35. Gender distribution did not differ significantly (p=0.102), although males were more common among HCMV-positive participants, representing 66.7% of TB cases and 88.9% of controls. Body mass index (BMI) categories differed significantly across groups (p<0.001). Underweight status was most common among HCMV-positive TB cases (58.3%), while healthy weight predominated among controls, particularly HCMV-positive controls (62.5%). HIV infection was present in 75.0% of HCMV-positive TB cases and 77.8% of HCMV-positive controls, compared with 29.1% of HCMV-negative TB cases and 33.7% of HCMV-negative controls (p=0.001) and the overall HIV prevalence was 105 (35.4%).

The proportion of PLWHIV with CD4 counts <200 cells/µl was similar between cases (10.2%) and controls (8.0%). Among the cases, 6 were anti-HCMV IgM positive (6.1%) and one had low anti-HCMV avidity (1.0%) (Table 2).

**Table 1. Demographic and clinical characteristics of study participants stratified by HCMV DNAemia results and tuberculosis disease status.**

| | Category | TB cases | | Controls | | All participants (n = 297) | P value |
|---|---|---|---|---|---|---|---|
| | | HCMV+ (n = 12) | HCMV- (n = 86) | HCMV+ (n = 09) | HCMV- (n = 190) | | |
| **Age group (%)** | 18-35 | 5 (41.7) | 48 (55.8) | 1 (11.1) | 72 (37.9) | 126 (42.4) | 0.005 |
| | 36-50 | 7 (58.3) | 25 (29.1) | 7 (77.8) | 73 (38.4) | 112 (37.7) | |
| | >50 | 0 (0.0) | 13 (15.1) | 1 (11.1) | 45 (23.7) | 59 (19.9) | |
| **Race (%)** | Black | 4 (33.3) | 28 (32.6) | 3 (33.3) | 68 (35.8) | 103 (34.7) | 0.962 |
| | Mixed ancestry | 8 (66.7) | 58 (67.4) | 6 (66.7) | 122 (64.2) | 194 (65.3) | |
| **Gender (%)** | Female | 4 (33.3) | 44 (51.2) | 1 (11.1) | 84 (44.2) | 133 (44.8) | 0.102 |
| | Male | 8 (66.7) | 42 (48.8) | 8 (88.9) | 106 (55.8) | 164 (55.2) | |
| **TB Cough Score (%)** | No | 0 (0.0) | 0 (0.0) | 0 (0.0) | 5 (2.6) | 5 (1.7) | 0.413 |
| | Yes | 12 (100.0) | 86 (100.0) | 9 (100.0) | 185 (97.4) | 292 (98.3) | |
| **Chest pain (%)** | No | 5 (41.7) | 25 (29.1) | 3 (33.3) | 43 (22.6) | 76 (25.6) | 0.343 |
| | Yes | 7 (58.3) | 61 (70.9) | 6 (66.7) | 147 (77.4) | 221 (74.4) | |
| **Respiratory rate (%)** | Bradypnea | 2 (16.7) | 7 (8.1) | 1 (11.1) | 39 (20.5) | 49 (16.5) | 0.032 |
| | Normal | 3 (25.0) | 24 (27.9) | 2 (22.2) | 70 (36.8) | 99 (33.3) | |
| | Tachypnea | 7 (58.3) | 55 (64.0) | 6 (66.7) | 81 (42.6) | 149 (50.2) | |
| **Currently smoking (%)** | No | 5 (41.7) | 32 (37.2) | 4 (44.4) | 71 (37.4) | 112 (37.7) | 0.965 |
| | Yes | 7 (58.3) | 54 (62.8) | 5 (55.6) | 119 (62.6) | 185 (62.3) | |
| **Previously smoking (%)** | No | 4 (80.0) | 16 (50.0) | 2 (50.0) | 48 (67.6) | 70 (23.6) | 0.279 |
| | Yes | 1 (20.0) | 16 (50.0) | 2 (50.0) | 23 (32.4) | 42 (14.1) | |
| **Previous TB disease (%)** | No | 8 (66.7) | 52 (60.5) | 3 (33.3) | 115 (60.5) | 178 (59.9) | 0.405 |
| | Yes | 4 (33.3) | 34 (39.5) | 6 (66.7) | 75 (39.5) | 119 (40.1) | |
| **HIV status** | Negative | 3 (25.0) | 61 (70.9) | 2 (22.2) | 126 (66.3) | 192 (64.6) | 0.001 |
| | Positive | 9 (75.0) | 25 (29.1) | 7 (77.8) | 64 (33.7) | 105 (35.4) | |
| **Body Mass Index category (%)** | Under weight | 7 (58.3) | 37 (43.0) | 1 (12.5) | 42 (22.1) | 87 (29.3) | <0.001 |
| | Healthy weight | 4 (33.3) | 44 (51.2) | 6 (62.5) | 95 (50.0) | 149 (50.2) | |
| | Overweight | 1 (8.3) | 5 (5.8) | 2 (25.0) | 53 (27.9) | 61 (20.5) | |

Data is presented as number and percentage (%) within each category. The reported p-values in the table are for comparison between cases (with tuberculosis) and controls (no tuberculosis) for each variable. The three Body Mass index categories have the following ranges; < 18.5, 18.5-24.9 and ≥25 for Underweight, Healthy weight and Overweight respectively. TB: tuberculosis. HCMV + : Participants with HCMV DNAemia. HCMV-: Participants without HCMV DNAemia

## Association of HCMV DNAemia with tuberculosis disease

In a univariate analysis, participants with TB had about three odds of HCMV DNAemia (odds ratio [OR] 2.95, 95% confidence interval [95%CI] 1.20–7.47; p = 0.019: Table 3). After multivariate adjustment for age, HIV status, gender and BMI, this association became stronger (adjusted OR [aOR] 4.99, 95%CI 1.63–16.99; p = 0.007).

The effect modification analysis showed no significant interaction between HCMV DNAemia and BMI category (interaction term p-values 0.297 and 0.298 for underweight and overweight, respectively; Table A in S1 Text).

In a stratified analysis of the HCMV DNAemia and TB disease by HIV status (Table 4), HCMV DNAemia was associated with higher odds of TB disease (OR: 3.20, 95%CI 1.26–8.41; p = 0.015), whereas HIV positivity was not significantly associated with TB disease (OR: 0.84, 95%CI 0.48–1.43; p = 0.53). In the interaction model, the effect of HCMV DNAemia on TB did not differ by HIV status.

When hemoglobin was included in the multivariable model as a covariate together with age, gender and HIV status, the association between TB status and HCMV DNAemia was attenuated and became statistically non-significant (aOR: 2.68, 95%CI 0.93–8.15; p = 0.073). Hence performing a mediation analysis where we found that hemoglobin partially mediated

**Table 2. Laboratory investigation results of study participants stratified by tuberculosis status.**

| | Category | TB Cases (n=98) | Controls (n=199) | All participants (n=297) | P value |
|---|---|---|---|---|---|
| **White Blood Cell count category (%)** | Leukocytosis | 32 (32.7) | 26 (13.1) | 58 (19.5) | <0.001 |
| | Leukopenia | 7 (7.1) | 24 (12.1) | 31 (10.4) | |
| | Normal | 58 (59.2) | 145 (72.9) | 203 (68.4) | |
| **Hemoglobin category (%)** | Anemic | 58 (59.2) | 60 (30.2) | 118 (39.7) | <0.001 |
| | Normal | 40 (40.8) | 139 (69.8) | 179 (60.3) | |
| **Platelet count category (%)** | Normal | 35 (35.7) | 155 (77.9) | 190 (64.0) | <0.001 |
| | Thrombocytopenia | 3 (3.1) | 6 (3.0) | 9 (3.0) | |
| | Thrombocytosis | 59 (60.2) | 34 (17.1) | 93 (31.3) | |
| **GeneXpert Ultra resistance (%)** | Indeterminate | 2 (2.0) | 0 (0.0) | 2 (0.7) | 0.933 |
| | Rif resistance detected | 10 (10.2) | 0 (0.0) | 10 (3.4) | |
| | Rif resistance not detected | 86 (87.8) | 1 (0.5) | 87 (29.3) | |
| **CD4 count (%)** | <200 cells/µl | 10 (10.2) | 16 (8.0) | 26 (8.8) | 0.656 |
| | >200 cells/µl | 23 (23.5) | 51 (25.6) | 74 (24.9) | |
| **Anti-HCMV IgM (%)** | Negative | 92 (93.9) | 186 (93.5) | 278 (93.6) | 1.000 |
| | Positive | 6 (6.1) | 13 (6.5) | 19 (6.4) | |
| **Anti-HCMV IgG avidity (%)** | High-avidity | 97 (99.0) | 198 (99.5) | 295 (99.3) | 1.000 |
| | Low-avidity | 1 (1.0) | 1 (0.5) | 2 (0.7) | |

Data is presented as number and percentage (%) within each category for cases (with tuberculosis) and controls (no tuberculosis), and the combined cohort. CD4 count: among participants with HIV infection only. Rif: rifampicin.

**Table 3. Logistic regression analysis of factors associating HCMV DNAemia with tuberculosis disease.**

| Factor | Category | Univariate analysis Crude OR (95%CI) | P-value | Multivariate analysis Adjusted OR (95%CI) | P-value |
|---|---|---|---|---|---|
| **HCMV DNAemia** | Negative | 1.00 (Reference) | | 1.00 (Reference) | |
| | Positive | 2.95 (1.20-7.47) | 0.019 | 4.99 (1.63-16.99) | 0.007 |
| **Age** | 18-35 | 1.00 (Reference) | | 1.00 (Reference) | |
| | 36-50 | 0.55 (0.32-0.94) | 0.031 | 0.66 (0.35-1.23) | 0.193 |
| | >50 | 0.39 (0.19-0.77) | 0.009 | 0.53 (0.24-1.10) | 0.094 |
| **Gender** | Female | 1.00 (Reference) | | 1.00 (Reference) | |
| | Male | 0.78 (0.48-1.26) | 0.308 | 0.47 (0.26-0.83) | 0.011 |
| **HIV status** | Negative | 1.00 (Reference) | | 1.00 (Reference) | |
| | Positive (CD4<200 cells/µl) | 0.25 (0.52-2.88) | 0.605 | 0.64 (0.22-1.75) | 0.391 |
| | Positive (CD4>200 cells/µl) | 0.90 (0.50-1.59) | 0.726 | 0.61 (0.30-1.18) | 0.149 |
| **BMI** | Healthy weight | 1.00 (Reference) | | 1.00 (Reference) | |
| | Underweight | 2.13 (1.24-3.68) | 0.006 | 2.28 (1.27-4.11) | 0.005 |
| | Overweight | 0.23 (0.08-0.53) | 0.001 | 0.18 (0.06-0.44) | <0.001 |

OR: Odds Ratio. BMI: Body Mass Index. The three BMI categories have the following ranges; <18.5, 18.5-24.9 and ≥25 for Underweight, Healthy weight and Overweight respectively.

**Table 4. Stratified analysis of HCMV DNAemia and TB disease by HIV status.**

| Model | Variable | OR (95% CI | P-value |
|---|---|---|---|
| **Adjusted** | HCMV DNAemia | 3.20 (1.26–8.41) | 0.015 |
| | HIV positivity | 0.84 (0.48–1.43) | 0.53 |
| **Interaction** | HCMV DNAemia | 3.15 (0.51–24.34) | 0.216 |
| | HIV positivity | 0.84 (0.47–1.46) | 0.539 |
| | HCMV DNAemia × HIV positivity interaction | 1.02 (0.10–8.52) | 0.984 |

OR: Odds Ratio.

the effect of HCMV DNAemia on TB disease, accounting for approximately one-fifth of the total effect (proportion mediated 0.21, p = 0.028), with a statistically significant mediated effect (Average Causal Mediation Effect [ACME] 0.069, p = 0.028) alongside a statistically significant direct effect (Average Direct Effect [ADE] 0.26, p = 0.010; Table B in S1 Text).

### Association of HCMV previous exposure with tuberculosis disease

Anti-HCMV IgM positivity was similar between cases (6.1%, n = 6) and controls (6.5%, n = 13). Only two participants had primary HCMV infection as indicated by low anti-HCMV IgG avidity: 1 case (1.0%) and 1 control (0.5%). Current HCMV infection was observed in 12 (12.2%) cases and 9 (4.5%) control, recent HCMV infection in 5 (5.1%) cases and 11 (5.5%) control, and Past HCMV infection in 80 cases (81.6%) and 178 controls (89.4%) (Table C in S1 Text).

The number of participants with primary HCMV infection was very small, therefore this category was excluded from further analyses. Participants with current HCMV infection had significantly higher odds of TB disease compared to those with past HCMV infection (crude OR 2.97, 95%CI 1.21–7.54, p = 0.018; aOR 4.99, 95%CI 1.62–16.73, p = 0.006) in a multivariable analysis that controlled for age, HIV status, gender and BMI as confounders (Table 5). Tuberculosis disease was not associated with recent HCMV infection (crude OR = 1.11, 95%CI 0.38–2.97; p = 0.837; aOR 0.99, 95%CI 0.27–3.17; p = 0.982) in either analysis.

**Table 5. Logistic regression analysis of factors associating HCMV previous exposure with tuberculosis disease.**

| Factor | Category | Univariate analysis Crude OR (95%CI) | P-value | Multivariable analysis Adjusted OR (95%CI) | P-value |
|---|---|---|---|---|---|
| **HCMV previous exposure** | Past HCMV infection | 1.00 (Reference) | | 1.00 (Reference) | |
| | Current HCMV infection | 2.97 (1.21-7.54) | 0.018 | 4.99 (1.62-16.73) | 0.006 |
| | Recent HCMV infection | 1.11 (0.38-2.97) | 0.837 | 0.985 (0.27-3.17) | 0.982 |
| **Age** | 18-35 | 1.00 (Reference) | | 1.00 (Reference) | |
| | 36-50 | 0.55 (0.32-0.94) | 0.031 | 0.66 (0.35-1.23) | 0.192 |
| | >50 | 0.39 (0.19-0.77) | 0.009 | 0.53 (0.24-1.10) | 0.094 |
| **Gender** | Female | 1.00 (Reference) | | 1.00 (Reference) | |
| | Male | 0.78 (0.48-1.26) | 0.308 | 0.46 (0.26-0.83) | 0.010 |
| **HIV status** | Negative | 1.00 (Reference) | | 1.00 (Reference) | |
| | Positive (CD4 < 200 cells/µl) | 0.25 (0.52-2.88) | 0.605 | 0.64 (0.22-1.75) | 0.394 |
| | Positive (CD4 > 200 cells/µl) | 0.90 (0.50-1.59) | 0.726 | 0.61 (0.30-1.18) | 0.149 |
| **BMI** | Healthy weight | 1.00 (Reference) | | 1.00 (Reference) | |
| | Underweight | 2.13 (1.24-3.68) | 0.006 | 2.28 (1.27-4.11) | 0.006 |
| | Overweight | 0.23 (0.08-0.53) | 0.001 | 0.18 (0.06-0.44) | <0.001 |

OR: Odds Ratio. BMI: Body Mass Index. The three BMI categories have the following ranges; < 18.5, 18.5-24.9 and ≥25 for Underweight, Healthy weight and Overweight respectively.

## Descriptive results of HCMV viral load across related TB disease history/status

The overall range of viral loads was 30.7–16,459.8 and 30.3–46,646.7 copies/ml among cases and controls respectively and the median (IQR) HCMV viral load of the respective groups was 88.9 (47.8–441.3) copies/ml and 107.4 (57.6–153.0) copies/ml. Higher HCMV viral loads were associated with longer time-to-positivity of the mycobacterial culture, i.e., with lower bacterial loads (Fig B in S1 Text). Among participants with HIV, cases generally had elevated HCMV viral loads compared to controls (Fig C in S1 Text). Furthermore, participants with a history of prior TB demonstrated greater variability in HCMV DNAemia levels, with some individuals exhibiting notably high viral loads, whereas those without prior TB disease mostly had low viral loads (Fig D in S1 Text). Cases with previous TB history displayed markedly elevated HCMV viral loads compared to TB cases without past TB history (Fig E in S1 Text).

## Discussion

In this cross-sectional analysis, 7.1% of patients diagnosed with TB had active HCMV infection based on HCMV DNAemia. After adjusting for confounding, participants with TB were five times more likely to have active HCMV infection than those without TB. Unlike recent infection, current HCMV infection was associated with increased odds of TB disease.

Our finding that current HCMV infection was strongly associated with tuberculosis in adults aligns with that from a South African longitudinal birth cohort study [5]. Unlike the Martinez *et al.* study which reported that active HCMV replication in infancy predicted later TB disease, our cross-sectional study explored the concurrent relationship between HCMV and TB among adults. Our findings show that current HCMV infection and TB may be associated when measured at the same time point but showed no association of active TB with earlier HCMV exposure (recent HCMV infection). These findings could suggest that while primary HCMV infection may influence TB susceptibility early in life, recent HCMV infection might play no or only a limited role in adult TB pathogenesis. The childhood and adulthood tuberculosis risk attributed to HCMV may be different because unlike adults in TB endemic countries who are likely to experience reactivation or reinfection with HCMV and Mtb, children are more likely to have primary diseases due to both pathogens.

HCMV repeatedly infects immunocompetent individuals [22–24] and these infections become persistent and incremental leading to superinfections with same or different genotypes in humans [25]. Persistent infection with HCMV leads to immunosenescence through immune inflation where there is depletion of naïve T cells and a tremendous expansion of effector memory CD8$^+$ T and CD4$^+$ T cells [26] due to chronic immune activation. Immune activation due to HCMV determined by IFNγ ELISPOT positivity in South African children was associated with a moderately high TB disease risk [9]. Children are expected to have less immune activation due to HCMV compared to adults who experience incremental exposure especially in endemic areas. A study done among Ugandan adults [27] associated HCMV specific IgG titers and inflammation which was measured by IP-10 and IL-1α with increased TB disease risk. HCMV reactivation or reinfection however may have a smaller role in immune activation in adults which may be due to chronic infections other than HCMV [28], age-related immunosenescence [29], and environmental agents and lifestyle factors [30], amongst others.

The observed association of current HCMV infection with TB as reported in this study may be due to expansion of the CD4$^+$CD28$^{null}$ T cells associated with impaired immune responses to numerous antigens which may include Mtb during the active virus replication phase [31]. This same study reported reduced proportion of this T cell subset during HCMV treatment with valacyclovir which may also indicate that during recent HCMV infection phase, there is lower HCMV antigens hence reduced CD4$^+$CD28$^{null}$ T cells and hence no reported association with TB disease in this group. Tuberculosis disease has a long preclinical phase with subclinical or incipient TB disease lasting for several months to years with a shorter progression period in HIV endemic settings [32]. The TB disease progression may be due to immune dynamics brought about by latent, reactivation and reinfection HCMV cycles over time rather than a single acute/sporadic effect. HCMV reactivation may also be a consequence of TB disease immune activation because the virus is mostly detected

from tissues undergoing inflammation [33]. An alternative hypothesis may be that underlying decline in cellular immune functionality with age [34] which may trigger both progression to TB disease and HCMV reactivation.

Our hypothesis of hemoglobin being a mediator of an HCMV DNAemia-driven susceptibility to TB disease may be true because HCMV infection has been indicated to have direct effects on the bone marrow leading to pancytopenia [35,36] as well as causing immune mediated hemolytic anemia [37,38]. Anemia particularly iron deficiency one has been associated with tuberculosis disease [39] through causing changes in T cell subsets and cytokine profiles [40]. The observed anemia may also be due to a reverse causation where it is being caused by either early or undiagnosed tuberculosis disease. Anemia has been a reported comorbid among tuberculosis patients [41–43] which has been attributed to inflammation [44,45] and there is evidence of reduced anemia with TB treatment without additional iron based interventions [46,47].

For the descriptive findings, higher HCMV viral loads were associated with longer mycobacterial culture time-to-positivity which may suggest that viral activity may correlate with lower bacterial burden. This contrasts with previous studies reporting a dose-response relationship between CMV exposure and TB risk [5,27]. Among participants with HIV, cases generally had elevated HCMV viral loads compared to controls, consistent with studies that have found impaired immune control of HCMV and increased viremia in people living with HIV [48,49]. The increased viral load in cases could be due to increased inflammatory environment in cases than in controls. Individuals with prior TB showed greater variability in HCMV DNAemia, with some cases exhibiting markedly high viral loads, possibly reflecting episodes of viral reactivation triggered by immune modulations. These observations suggest that HCMV reactivation dynamics may be influenced by both host immune status and TB disease history.

Our study had several limitations that could affect interpretation and generalization of the findings. Due to the cross-sectional design, we could not exclude with certainty that HCMV exposure preceded TB disease. We could not control for potential residual confounders like immunological status (beyond HIV), socio-economic status, behavioral factors such as alcohol use, co-infections, comorbidities, and host genetics. The use of plasma in diagnosing HCMV DNAemia was reported to miss out some positive participants especially those that had low DNAemia in whole blood [50] hence we could have missed some participants with latent or low-level reactivation. The Euroimmun HCMV specific IgM ELISA kits were reported to have a low sensitivity for serum samples stored beyond 7 weeks [51] and hence we could have misclassified HCMV exposure categories. Under HCMV exposure categories in secondary analysis, some categories had very small numbers of participants, which limits interpretability and statistical robustness. The high proportion of individuals with history of previous TB treatment and those who are of mixed-race ancestry may affect the generalization of our results to other settings.

In conclusion, our findings show that TB disease was strongly associated with current active HCMV infection but not with recent HCMV infection. This may indicate that among adults in high HCMV and TB incidence settings, HCMV replication may impair immune responses to other infections, active TB disease could be reactivating HCMV replication or there could be an underlying factor triggering both infections. Prospective studies carried out in multiple settings are needed to determine the direction of causality as well as accounting for residual confounders.

## Supporting information

**S1 Data. Data file.**
(XLSX)

**S1 Text. Supplementary description, tables and figures.**
(DOCX)

## Acknowledgments

The authors would like to thank BAR-TB Dx participants and the study team for their contribution to this study.

## Author contributions

**Conceptualization:** Derrick Semugenze, Achilles Katamba, Alberto García-Basteiro, Moses L Joloba, Frank Cobelens, Willy Ssengooba.

**Data curation:** Derrick Semugenze, Arthur Chiwaya, Alberto García-Basteiro, Moses L Joloba, Grant Theron, Frank Cobelens, Willy Ssengooba.

**Formal analysis:** Derrick Semugenze, Arthur Chiwaya, George William Kasule, James Sserubiri, Rose Nabatanzi, Achilles Katamba, Alberto García-Basteiro, Moses L Joloba, Grant Theron, Frank Cobelens, Willy Ssengooba.

**Investigation:** Derrick Semugenze, Arthur Chiwaya, George William Kasule, James Sserubiri, Rose Nabatanzi, Achilles Katamba, Alberto García-Basteiro, Moses L Joloba, Grant Theron, Frank Cobelens, Willy Ssengooba.

**Methodology:** Derrick Semugenze, Arthur Chiwaya, George William Kasule, James Sserubiri, Rose Nabatanzi, Byron W P Reeve, Zaida Palmer, Hridesh Mishra, Achilles Katamba, Alberto García-Basteiro, Moses L Joloba, Grant Theron, Frank Cobelens, Willy Ssengooba.

**Supervision:** Alberto García-Basteiro, Moses L Joloba, Frank Cobelens, Willy Ssengooba.

**Writing – original draft:** Derrick Semugenze, Alberto García-Basteiro, Moses L Joloba, Frank Cobelens, Willy Ssengooba.

**Writing – review & editing:** Derrick Semugenze, Arthur Chiwaya, George William Kasule, James Sserubiri, Rose Nabatanzi, Zaida Palmer, Hridesh Mishra, Achilles Katamba, Alberto García-Basteiro, Moses L Joloba, Grant Theron, Frank Cobelens, Willy Ssengooba.

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
