## [Decision Letter · Decision Letter 0]

3 Mar 2026

PGPH-D-26-00309

Association of Human Cytomegalovirus exposure with tuberculosis disease in South African adults with presumptive tuberculosis

Dear Dr. Semugenze,

Thank you for submitting your manuscript to PLOS Global Public Health. After careful consideration, we feel that it has merit but does not fully meet PLOS Global Public Health’s publication criteria as it currently stands. Therefore, we invite you to submit a revised version of the manuscript that addresses the points raised during the review process.

We look forward to receiving your revised manuscript.

Kind regards,

Leonardo Martinez

Academic Editor

Journal Requirements:

Additional Editor Comments (if provided):

Reviewers' comments:

Reviewer's Responses to Questions

**Comments to the Author**

1. Does this manuscript meet PLOS Global Public Health’s publication criteria? Is the manuscript technically sound, and do the data support the conclusions? The manuscript must describe methodologically and ethically rigorous research with conclusions that are appropriately drawn based on the data presented.

Reviewer #1: Partly

Reviewer #2: Yes

Reviewer #3: Yes

2. Has the statistical analysis been performed appropriately and rigorously?

Reviewer #1: Yes

Reviewer #2: No

Reviewer #3: Yes

3. Have the authors made all data underlying the findings in their manuscript fully available (please refer to the Data Availability Statement at the start of the manuscript PDF file)?

Reviewer #1: Yes

Reviewer #2: Yes

Reviewer #3: Yes

4. Is the manuscript presented in an intelligible fashion and written in standard English?

Reviewer #1: Yes

Reviewer #2: Yes

Reviewer #3: Yes

5. Review Comments to the Author

Reviewer #1: Thank you for the opportunity to review this work. The research question is relevant and timely. My primary concern relates to the categorization of HCMV exposure in the secondary analyses. The grouping strategy is difficult to follow and, in some instances, combines physiopathologically distinct entities (e.g., primary acute CMV infection with CMV reactivation or reinfection). Furthermore, some categories include very small numbers of participants, which limits interpretability and statistical robustness.

If possible, I suggest simplifying the nomenclature and analytical framework to improve clarity and readability. For example, instead of using the category “current HCMV reactivation or reinfection” consider using “current HCMV infection” with clear explanation in Methods that “current” infection may include new infection, reactivation or reinfection. A similar simplification could be applied to the “recent” categories (e.g. “recent CMV infection” instead of “recent CMV infection or reinfection”).

Below are my specific comments:

ABSTRACT:

1. Lines 33-37: The sentence contains minor grammatical errors and unclear phrasing, particularly regarding the definition of cases and controls. Please revise for clarity and precision.

2. Lines 40-42: Revise. The main results presented in an Abstract should prioritize the primary analysis, such as providing and comparing the proportion of participants positive CMV DNAemia in both study arms.

3. Line 42-44: Again, why are the results from the logistic regression for the secondary outcome (combined HCMV serology and DNA results) presented instead of the primary outcome (DNA only)?

4. Line 43-47: The multiple HCMV categories are difficult to follow in the Abstract and require readers to consult Methods section for interpretation. I suggest simplifying this classification.

INTRODUCTION:

1. Lines 71-72: The statement regarding the evidence for the existence of CMV reinfections (with a genetically distinct strain form the original), does not appear to be directly supported by reference n°10. Please verify and revise the citation accordingly.

METHODS:

1. Lines 82-83: Please clarify:

• Who are the participants with “non-actionable culture results” ?

• Who are “those with missing” (missing what?)

• What is meant by “inappropriate” sample? or “missing sample volumes”?

2. Line 84-87 (Controls):

Please clearly define the control group:

• Were they individuals with presumptive pulmonary TB in whom the diagnosis was ultimately excluded?

• How were controls selected? The manuscript states they were selected “randomly”, but the procedure requires detailed description to ensure internal validity.

3. Lines 92-98 (HCMV qPCR):

• Please provide a reference for real-time qPCR method.

• Was qPCR performed in serum or plasma?

• Clarify that the primary outcome (HCMV DNAemia) was analyzed qualitatively (positive/ negative)

4. IgG avidity testing:

Please provide the rationale for including HCMV IgG avidity testing. Describe its temporal interpretation in HCMV infection and cite references specific to HCMV IgG avidity.

5. CMV Exposure Categories (Box):

The exposure categories require stronger justification, and ideally supporting references. Each category should be clearly and consistently defined:

• “Primary CMV Infection”: Given that IgM may persist positive for up to 6 months (Lazzarotto T, et al. J Clin Virol. 2008;41(3):192–197), on what basis were participants with negative IgM but positive IgG with low avidity classified as “primary infection”? Also, how were participants with negative IgG but positive IgM classified?

• “Recent CMV reactivation or reinfection”: Why was a positive IgM required for this category? Many reinfections and reactivations occur without detectable IgM.

• “Past CMV infection”: revise the definition, there appears to be a repetition in the final sentence.

6. Lines 142-146: Are these considered secondary outcomes or exploratory analyses? Please clarify.

7. Line 144: Please clarify the outcome described as “variation according to previous TB history”.

8. Sample size: Please provide sample size calculations for the primary outcome, including assumed effect size, power, alpha, and the intended case-control ratio.

RESULTS:

1. Characteristics of participants with positive HCMV DNAemia:

• Please describe the demographic and clinical characteristics (age, sex, HIV status, etc) among CMV DNApositive participants in both study arms.

• Given the known interaction between HIV and CMV reactivation, I strongly recommend adding a stratified analysis of the primary outcome (HCMV DNAemia and TB disease) by HIV status.

2. Line 171: Please provide the proportion of HCMV DNA-positive participants for both groups, not only among TB cases.

3. Tables 1 and 2: Provide p-values for baseline group comparisons (including those mentioned in lines 161-169). Also, please present variables consistently (either Yes/No, or Present/Absent, but not both).

4. Secondary results (from line 185 onward): Please organize results according to the same infection categories presented in the Box and maintain consistent nomenclature throughout.

5. Merging of exposure categories:

The category “primary HCMV infection” (N=1 per group) was merged with “recent HCMV reactivation or reinfection,” creating a new category (“recent HCMV infection, reactivation or reinfection”). This is problematic for two reasons:

• These conditions are physiopathologically distinct.

• The resulting category is difficult to interpret.

I would recommend excluding the primary infection category from analysis rather than merging it.

6. Supplementary. Figures 3 and 4:

Please provide statistical significance directly in the figures.

DISCUSSION:

1. Low hemoglobin as mediator:

The hypothesis that anemia may mediate the relationship between CMV and TB is interesting. However, reverse causality should be discussed. Anemia may reflect early, undiagnosed TB rather than a mediator of CMV-driven susceptibility.

2. Biological sampling limitations:

Please discuss limitations of measuring HCMV DNAemia in plasma versus whole blood, particularly regarding sensitivity for detecting latent or low-level reactivation.

3. Lines 216-218: The sentence is difficult to follow and requires revision: “Current HCMV reactivation or reinfection was associated with increased odds of TB disease well as recent HCMV infection, reactivation or reinfection was not.”

4. Interpretation of secondary subgroup analyses: The study appears underpowered for multiple subgroup comparisons. Therefore, statements such as (line 223-227) “These findings suggest that […] recent HCMV infection, reactivation or reinfection plays no or only a limited role in adult TB pathogenesis” remain speculative and should be tempered.

5. Temporal argument against causality (Lines 244–252): TB has a prolonged preclinical phase that may extend over several months. The finding that only current—but not recent or prior—CMV reactivation/reinfection is associated with TB disease also weakens a causal interpretation. This issue should be discussed more critically.

Reviewer #2: Dear Authors

Thank you for your submission. Quite an interesting article Please see below some comments.

Introduction

1. Please update to 2024 numbers

Methods

2. So both cases and controls have presumptive TB with symptoms? Please clarify

3. Seems both a change in estimate and DAG were used for confounder assessment. Can you please explain the logic behind using both especially when you have a DAG?

Results

4. Line 165. There is a type. N=297 is overall and not the n for HIV

5. Line 171. Not sure what the N and % here are referring to. Please clarify

6. Can you explain the hypothesized pathway for why Hg would be a mediator in this association? How does HCMV lead to change in Hg that then leads to TB?

7. Why was primary merged with recent reactivation/reinfection and not current infection? One would think primary infection would be similar to current re-infection.

Discussion

8. I would refrain from calling it nested CC. Nested CC may refer to a case control study in a prospective cohort with longitudinal follow up where exposure is collected before the outcome. Here you are doing a cross-sectional analysis with both being assessed at the same time.

9. Line 253. This is an interesting finding. If the hypothesis is that TB leads to viral reactivation then low burden of TB bacteria should show low viremia and a dose response (as TB burden increases so does viremia) or higher burden of TB with low viremia. However if viremia leads to TB reactivation, then higher burden of viremia can lead to TB disease even with smaller TB load.

Figure 1

10. Not sure what randomly selected means as 98 of 99 were selected.

11. Box. "and had high anti-HCMV IgG avidity" is repeated for Past HCMV infection definition.

Reviewer #3: Significance: This study by Semugenze, et al. aims to define the association between recent or active cytomegalovirus (CMV) infection and TB disease. Several prior studies have documented an association between acquired CMV infection and TB risk, particularly in pediatric populations. In adults, the majority of the population that is also susceptible to TB is CMV seropositive, but CMV reactivation may still impact TB disease progression. Controls were matched to TB cases at 2:1 in this South African cohort, which were divided based on recency of CMV infection into: primary HCMV infection, current CMV reactivation/infection, recent CMV reactivation/re-infection and past CMV infection using a combination of qPCR and antibody tests. Current CMV reactivation showed a strong association with TB, which was supported by the data. Overall, this is an important study that strengthens the hypothesis that CMV infection increases TB risk.

Minor comments:

- Can you explain the rationale of combining the two people with primary HCMV infection (one TB case and one control) with the rest of the HCMV reactivation/re-infection group? It would be a better sensitivity analysis to remove these members, especially at such a small sample size.

- Is there any support in the literature for the idea that TB itself is driving CMV reactivation?

- Can you speculate reasons for the association between HCMV and longer time to positivity? Are you suggesting that CMV-induced inflammation is associated with faster Mtb control? This seems counterintuitive based on the rest of the data in the paper.

6. PLOS authors have the option to publish the peer review history of their article (what does this mean?). If published, this will include your full peer review and any attached files.

**Do you want your identity to be public for this peer review?** For information about this choice, including consent withdrawal, please see our Privacy Policy.

Reviewer #1: No

Reviewer #2: No

Reviewer #3: No

Figure Resubmissions:

---

## [Decision Letter · Decision Letter 1]

8 Apr 2026

PGPH-D-26-00309R1

Association of Human Cytomegalovirus exposure with tuberculosis disease in South African adults with presumptive tuberculosis

Dear Dr. Semugenze,

Thank you for submitting your manuscript to PLOS Global Public Health. After careful consideration, we feel that it has merit but does not fully meet PLOS Global Public Health’s publication criteria as it currently stands. Therefore, we invite you to submit a revised version of the manuscript that addresses the points raised during the review process.

We look forward to receiving your revised manuscript.

Kind regards,

Leonardo Martinez

Academic Editor

**Journal Requirements:**

**Reviewers' comments:**

Reviewer's Responses to Questions

**Comments to the Author**

1. If the authors have adequately addressed your comments raised in a previous round of review and you feel that this manuscript is now acceptable for publication, you may indicate that here to bypass the “Comments to the Author” section, enter your conflict of interest statement in the “Confidential to Editor” section, and submit your "Accept" recommendation.

Reviewer #1: All comments have been addressed

Reviewer #2: All comments have been addressed

Reviewer #3: All comments have been addressed

2. Does this manuscript meet PLOS Global Public Health’s publication criteria? Is the manuscript technically sound, and do the data support the conclusions? The manuscript must describe methodologically and ethically rigorous research with conclusions that are appropriately drawn based on the data presented.

Reviewer #1: Yes

Reviewer #2: Yes

Reviewer #3: Yes

3. Has the statistical analysis been performed appropriately and rigorously?

Reviewer #1: Yes

Reviewer #2: Yes

Reviewer #3: Yes

4. Have the authors made all data underlying the findings in their manuscript fully available (please refer to the Data Availability Statement at the start of the manuscript PDF file)?

Reviewer #1: Yes

Reviewer #2: Yes

Reviewer #3: Yes

5. Is the manuscript presented in an intelligible fashion and written in standard English?

Reviewer #1: No

Reviewer #2: Yes

Reviewer #3: Yes

6. Review Comments to the Author

**Reviewer #1:** The manuscript has substantially improved following revision. However, a few minor issues remain that should be addressed for clarity and consistency.

ABSTRACT

• - The sentence spanning lines 37–39 is unintelligible (“…was detected by qPCR well as…”). The word “well” appears to be a typographical error. Beyond this, the sentence structure is unclear and should be rewritten for readability. For clarity, the authors may consider explicitly defining HCMV infection categories using parentheses. For example: “Current HCMV infection (including reactivation or reinfection) and recent infection were classified using…”.

• - The sentence spanning lines 41–43 is also unclear: “A similar association was observed for current HCMV infection well as no association with recent HCMV infection was found.” The structure is confusing and difficult to interpret. Please revise for clarity.

METHODS:

• - Primary HCMV infection is defined (line 125) as a combination of positive anti-HCMV IgM and low IgG avidity. However, the working definition provided in Box 1 allows classification of primary infection in the absence of IgM positivity. The rationale should be explained in Methods.

RESULTS:

• - Lines 217–218: Please revise the following sentence for clarity: “Comparable proportions between cases and controls were observed in anti-HCMV IgM results with 6.1% (n=6) and 6.5% (n=13) being positive, respectively.” The sentence is difficult to read and could be simplified.

• - Line 218: Replace “2” with “two”.

• - Table 1: Given the multiple columns presented, please specify (e.g., in a footnote) what the reported p-values refer to (e.g., comparison between TB cases vs. non-TB controls for each variable).

• - Table 2 (first row headers): Replace “Cases” with “TB cases” for clarity.

**Reviewer #2:** Thank you for the edits. I do not have any more major comments.

One minor thing would be to restrict the ORs and 95% CIs to 2 significant figures given limited sample size

**Reviewer #3:** The authors have adequately addressed the comments

7. PLOS authors have the option to publish the peer review history of their article (what does this mean?). If published, this will include your full peer review and any attached files.

**Do you want your identity to be public for this peer review?** For information about this choice, including consent withdrawal, please see our Privacy Policy.

Reviewer #1: No

Reviewer #2: No

Reviewer #3: **Yes**: Sara Suliman

**Figure Resubmissions:**

---

## [Editor Report · Decision Letter 2]

13 Apr 2026

Association of Human Cytomegalovirus exposure with tuberculosis disease in South African adults with presumptive tuberculosis

PGPH-D-26-00309R2

Dear Semugenze,

We are pleased to inform you that your manuscript 'Association of Human Cytomegalovirus exposure with tuberculosis disease in South African adults with presumptive tuberculosis' has been provisionally accepted for publication in PLOS Global Public Health.

Best regards,

Leonardo Martinez

Academic Editor
